# Octopus-Inspired Soft Robot for Slow Drug Release

**DOI:** 10.3390/biomimetics9060340

**Published:** 2024-06-04

**Authors:** Dingwen Tong, Yiqun Zhao, Zhengnan Wu, Yutan Chen, Xinmiao Xu, Qinkai Chen, Xinjian Fan, Zhan Yang

**Affiliations:** 1School of Mechanical and Electrical Engineering, Soochow University, Suzhou 215131, China; 2129401038@stu.suda.edu.cn (D.T.); 2129401067@stu.suda.edu.cn (Y.Z.); 20225229110@stu.suda.edu.cn (Z.W.); chenqinkai0426@163.com (Q.C.); 2School of Future Science and Engineering, Soochow University, Suzhou 215222, China2262402039@stu.suda.edu.cn (X.X.)

**Keywords:** multimodal locomotion, soft robot, octopus-inspired, suction cup, drug delivery, magnetic actuation

## Abstract

Octopus tentacles are equipped with numerous suckers, wherein the muscles contract and expel air, creating a pressure difference. Subsequently, when the muscular tension is released, objects can be securely adhered to. This mechanism has been widely employed in the development of adhesive systems. However, most existing octopus-inspired structures are passive and static, lacking dynamic and controllable adhesive switching capabilities and excellent locomotion performance. Here, we present an octopus-inspired soft robot (OISR). Attracted by the magnetic gradient field, the suction cup structure inside the OISR can generate a strong adsorption force, producing dynamically controllable adsorption and separation in the gastrointestinal (GI) tract. The experimental results show that the OISR has a variety of controllable locomotion behaviors, including quick scrolling and rolling motions, generating fast locomotion responses, rolling over gastric folds, and tumbling and swimming inside liquids. By carrying drugs that are absorbable by GI epithelial cells to target areas, the OISR enables continuous drug delivery at lesions or inflamed regions of the GI tract. This research may be a potential approach for achieving localized slow drug release within the GI tract.

## 1. Introduction

After billions of years of natural selection, nature has given birth to invertebrate organisms such as octopuses, which possess powerful adhesive capabilities. Octopuses can generate significant adsorption on different surfaces through the suction-cup-like structures on their tentacles. In recent years, bio-inspired flexible adhesive structures based on octopus tentacle suckers have become a research hotspot in various fields, including adhesive mechanism studies [1], electrophysiological signal acquisition [2,3,4], wearable hydrogels [5], wound healing management [6], and high-performance soft grippers [7,8,9]. Due to the strong resistance to detachment exhibited by octopus tentacle suckers, incorporating the suction cup structure into a patch form has demonstrated excellent mechanical performance [10]. Moon Kee Choi et al. developed state-of-the-art nanostructured biomimetic miniature suckers (mSCs) that can be used for continuous monitoring of vital signs in living organisms and enable repeated drug reloading without loss of adsorption strength [11]. Sangyul Baik et al. developed an artificial reversible wet/dry adhesive octopus-inspired architectures (OIAs) that exhibits potent, reversible, and highly repeatable adsorption to silicon wafers, glass, and rough skin surfaces under various conditions [10]. Jihyun Lee et al. developed a millimeter-scale dual-layer suction cup cluster (d-SCC) non-invasive transdermal patch that allows nano-scale deformation control of the stratum corneum in contact with the skin, enabling effective transport of various drugs through the stratum corneum without causing skin discomfort [12]. Suhao Wang et al. developed a centimeter-scale octopus-inspired magnetically controlled adhesive that mimics both the morphology and muscle-driven motion of octopus suckers, providing a fast, adjustable, and highly reversible adsorption solution with a wide range of switching and adsorption strengths [13]. Inspired by the unique structural features of octopus suckers, Zhi Luo et al. designed a self-adaptive centimeter-scale suction patch that can exert powerful adsorption and effective mechanical deformation on mucosal tissues [14]. However, most of the developed bio-inspired octopus sucker structures are currently static and passive, are mainly used for surface adsorption on the skin and ex vivo drug release, and have limited switching capabilities. The development of octopus-inspired adhesive structures still faces challenges to achieving controllable and continuous targeted drug delivery tasks in vivo [15].

Soft active materials exhibit excellent mechanical functionality that surpasses many active components manufactured from rigid materials, leading to widespread application [16,17,18,19]. Among these materials, magnetic soft materials are particularly intriguing because they can be remotely controlled and rapidly manipulated in different media, as magnetic fields can penetrate a wide range of environments without causing interference [20,21,22,23,24,25,26,27,28]. These magnetic soft materials are typically composed of hard magnetic particles or discrete magnets, where the desired magnetization mode direction and magnitude can be controlled by orienting the magnetization of disordered magnetic domains [29,30] or by applying a magnetic field during fabrication [31,32]. For the treatment of many GI disorders, the current mainstream approach involves orally administering drugs that dissolve and are absorbed in the GI system. This method requires the ingestion of large doses of drugs to achieve the desired therapeutic effect [33,34,35]. This not only results in drug wastage but also burdens patients with drug metabolism [36,37]. Several magnetic soft robots with drug-carrying capabilities have been developed to achieve targeted drug delivery, thereby reducing the intake of drug doses. Qiji Ze et al. developed a magnetically actuated amphibious folding soft robot that integrated multi-modal motion with rotational capability, liquid drug delivery, cargo transportation, and wireless operation [38]. Yue Dong et al. constructed soft robots with programmable magnetization curves and geometries by embedding programmed magnetization modules into adhesive sticker layers and integrating multiple functional modules [39]. Zhicheng Ye et al. developed a liquid-metal soft electronic coupled multi-legged robot that could be remotely moved and sensed through coupled magnetic and radiofrequency effects for targeted delivery in the GI tract [40]. These robots possess high degrees of maneuverability and motion capabilities [41,42]. However, current magnetic soft robots mostly deliver drugs to specific locations and cannot achieve localized slow drug release at the site of the lesion [23]. However, many GI disorders require slow drug release (continuous drug delivery) to achieve therapeutic efficacy. For ulcers in the stomach or duodenum, slow drug release can help control acidity and promote healing [43]. For inflammatory bowel disease, slow drug release can effectively restore intestinal microbiota diversity [44]. Therefore, developing a magnetic soft robot with a slow drug-release mechanism holds significant importance.

This study aims to develop an octopus-inspired soft robot (OISR) that can achieve slow drug release inside the body for therapeutic purposes. As shown in Figure 1a, we have taken gastric administration as an example; the OISR can be encapsulated within a capsule shell and introduced into the body. Guided by an external magnetic field, the OISR can release near the target location in the stomach and navigate to the lesion site through quick scrolling and rolling motions. The biomimetic suction cup of the OISR serves as a drug reservoir. By the transient attraction of magnetic gradient forces, the OISR can stably adhere to and remain on the surface of gastric tissue. Furthermore, the drug within the suction cup can be slowly released at the site of the lesion, thus achieving targeted slow drug release. As shown in Figure 1b, the OISR can perform quick scrolling and rolling motions by applying different forms of external magnetic fields. Quick scrolling includes instantaneous scrolling and following scrolling. Rolling can be divided into flat rolling, anti-gravity rolling, tumble climbing, and tumble swimming. The diverse motion modes enable the OISR to adapt to different external environments. Adsorption and separation can be achieved by applying a magnetic gradient field to the OISR. The intensity of the applied magnetic gradient influences the level of the adsorption force. In the paper, we introduce the design and fabrication method of the OISR, analyze and evaluate the implementation methods and motion performance of its different motion modes, and demonstrate the guided delivery and release process inside the body. Finally, we store simulated drugs within the suction cup of the OISR and control its motion to the simulated delivery site for adsorption and localized slow drug release. Additionally, leveraging its strong magnetism, the OISR can easily adhere to and remove magnetizable foreign bodies mistakenly swallowed by infants, which may have significant clinical application potential.

## 2. Materials and Methods

### 2.1. Design of OISR

The overall structure of the OISR consists of two parts: the biomimetic octopus suction cup and the robot body, which is capable of multi-modal motion. As shown in Figure 2a, to reduce the thickness of the OISR and minimize the impact of the suction cup structure on its motion performance, the suction cup is placed within the concave-shaped structure. During motion, the suction cup remains within its concave-shaped structure. Under the influence of a strong magnetic gradient field, it can be depressed to achieve adsorption. Additionally, to facilitate the depression motion of the suction cup, it is necessary to disrupt the dome-like structure of the concave surface, as the dome-like structure would lead to unstable force distribution states [45] and significantly hinder the depression motion. The concave-shaped structure is deformed into six gripping hooks, enabling the OISR to perform a tumble climbing motion and easily traverse the folds within the stomach. The gripping hooks should not possess magnetism since they tend to come into contact with each other during motion, and if they were magnetic, they would attract each other. To enhance the motion stability of the OISR, we have thickened the edge structures and designed wheel-like structures. With the widened edge portions, the center of gravity of the OISR can maintain greater stability when carrying drugs. As shown in Figure 2b, the biomimetic octopus suction cup structure is similar to the suction cups found on octopus tentacles. It possesses three key elements: a constricted orifice, a disk-like infundibulum, and a dome-shaped acetabulum. The acetabulum is a dome-shaped structure that generates low-pressure force and can be injected with different drug formulations. The disk-like infundibulum provides flexibility for adapting to various substrates. The constricted orifice prevents leakage and maintains suction [14]. Based on these characteristics of the octopus suction cup, we designed the biomimetic suction cup structure and added a gradient-force-enhancing unit. The suction cup can generate pre-tightening action at the target location by applying a strong gradient magnetic field to the OISR. After removing the gradient magnetic field, a stable pressure difference is maintained between the inside and outside of the suction cup, maintaining a certain suction level. In this way, the drug stored inside the suction cups can be continuously delivered on the surface of the GI.

### 2.2. Preparation of OISR

We use photosensitive resins with higher hardness to prepare the mold (Tough 2000 Resin, Formlabs, Somerville, MA, USA). The preparation process, dimensions, and partitioning of the mold are shown in Figure 2c. The OISR was fabricated using a stepwise filling method, which involved sequentially filling the suction cup, grappling hook, and center channel and joints. To adapt to different surface shapes, the sucker portion needed to be relatively soft and possess a certain level of resilience. Therefore, silicone rubber (Ecoflex™ 00-30, Smooth-On, Macungie, PA, USA) was chosen for filling this part. The intermediate groove and connection areas were primarily filled with magnetic soft materials, which, under the influence of a gradient magnetic field, allowed the suckers to generate downward pressure. However, due to the localized stiffness loss caused by the destruction of the dome structure, silicone rubber (Ecoflex™ 00-50, Smooth-On, Macungie, PA, USA) mixed with NdFeB particles (MQFP-15-7, Magnequench, Toronto, Ontario, Canada) was used for filling. The purpose was to partially reduce the stiffness reduction at the connection area, ensuring that the OISR could maintain stability while quickly scrolling. The grappling hook portion needed to have a certain degree of hardness and low viscosity to maintain the overall shape of the OISR better and avoid adsorption between the hook and sucker. Therefore, a mixture of silicone rubber (Ecoflex™ 00-50, Smooth-On, Macungie, PA, USA) and higher-hardness PDMS (Sylgard 184, Farnell, Leeds, UK) was used for filling. Due to the small size of the OISR, it was necessary to ensure that the mold surface was as clean as possible and free from surface deposits. A silicone rubber mold release agent (Ease Release^®^ 200, Smooth-On, Macungie, PA, USA) was applied in advance. As shown in Figure 2d, after mixing Ecoflex™ 00-30 (weight ratio A:B = 1:1) and vacuuming for 5 min, the mixture was injected into the mold for the sucker portion. Then, a miniature permanent magnet (N45) with a bottom diameter of 3 mm and a height of 1 mm was placed in the pre-made groove as a gradient force enhancement unit. The detailed assembly process of this part is shown in Appendix A. Subsequently, the mold was vacuumed for 10 min using a vacuum pump. The mold was then placed in a thermal oven and heated at 80 °C for 20 min. Next, the PDMS monomer (Part A) and cross-linking agent (Part B) were mixed in a volume ratio of 10:1. After mixing with Ecoflex™ 00-50 (weight ratio A:B = 1:1), the two mixtures were further combined in a weight ratio of 1:3. Vacuuming was performed for 5 min, followed by injection into the mold for the hook portion. Furthermore, Ecoflex™ 00-50 (weight ratio A:B = 1:1) was mixed with NdFeB particles, stirred evenly, vacuumed for 5 min, and then injected into the mold for the center channel and joint area. Finally, the mold was placed in a thermal oven and heated at 70 °C for 60 min.

After the fabrication of the OISR, it underwent magnetization treatment to ensure the presence of a stable magnetic moment, enabling it to be magnetized when exposed to a magnetic field. The magnetization intensity is influenced by the external magnetic field, and the relationship between the magnetic moment of the robot after magnetization $m_{0}$ and the intensity of the external magnetic field **H** can be expressed as follows [46]:(1)$$m_{0}=\chi_{\alpha }H$$
where $\chi_{\alpha }$ represents the magnetic susceptibility tensor of the material. As shown in Figure 2e, the OISR’s plane is attached to a 3D-printed fixture and placed in a capacitive pulse power supply (J1801N, JinL, Nanjing, China) for magnetization under a pulse magnetic field of approximately 1.8 T. The magnetization direction should align with the magnetization direction of the gradient-force-enhancing unit in the sucker. After the magnetization process is completed, the high coercivity of the NdFeB particles results in the OISR exhibiting a stable magnetic moment. Using a Tesla meter (F-30, CH-Magnetoelectricity Technology, China), we measured the distribution of the surface magnetic flux density of the OISR at a height of 0.5 mm, which is the closest distance to the OISR surface when the gradient-force-enhancing unit is not assembled. As shown in Figure 2f, the concave side has a magnetic pole labeled as “S” (magnetic induction line inflow) with a maximum magnetic flux of approximately 8 mT and a minimum magnetic flux of approximately 4 mT. As shown in Figure 2g, the plane side has a magnetic pole labeled as “N” (magnetic induction line outflow) with a maximum magnetic flux of approximately 17 mT and a minimum magnetic flux of approximately 8 mT. Additionally, the external magnetic fields of the six grappling hooks exhibit good distribution consistency. This uniform magnetic field distribution benefits the stable force exertion on the OISR.

The magnitude of the magnetic moment of the OISR directly affects its motion performance. Therefore, it is necessary to set the magnetic moment of the OISR appropriately to control its motion better. To characterize the dispersions of different contents of magnetic particles in the elastomer, NdFeB particles with mass fractions of 25%, 30%, 40%, 50%, 60%, and 70% were added to well-mixed Ecoflex™ 00-50 (weight ratio A:B = 1:1), which was vacuumed after thorough stirring. A syringe was used to draw 2 mL of the mixture, which was then dropped onto the center of a circular glass slide and spin-coated at 250 r/s for 90 s until a transparent thin film was obtained. The glass slide with the thin film was placed on a heating stage and cured at a constant temperature of 80 °C for 20 min. Subsequently, the film was observed under an optical microscope (NE950, Nexcope, Ningbo, China). The dispersions of magnetic particles in the elastomer at different ratios are shown in Appendix A. Although a higher proportion of magnetic particles results in a more significant magnetic moment of the OISR, which is advantageous for its motion, an excessive amount of magnetic particles significantly increases the Young’s modulus of the OISR, which is unfavorable for generating significant deformation [47]. Moreover, a high concentration of the premixed elastomer with the magnetic particles hinders proper material filling. To ensure that the OISR has better deformability, we ultimately chose NdFeB particles with a mass fraction of 60% as the preparation scheme.

### 2.3. Principles of Motion and Adsorption in OISR

Assuming the robot is to be controlled is a permanent magnet with a magnetic moment represented by **m** and located at position **p**, the torque **T** exerted on the robot when an external magnetic field with a magnetic flux density of **B**(**p**) is applied at point **p** can be expressed as follows:(2)T=m×B(p)

The torque **T** possesses the ability to align the magnetic moment of the robot with the external magnetic field by inducing rotational motion, as illustrated in Figure 3a(i). This is why the OISR stays “standing” when quickly scrolling.

Due to the non-uniform magnetic field generated by the permanent-magnet driver, which decreases continuously with increasing distance, the force **F** exerted on the robot can be calculated as follows:(3)$$F=\frac{3\mu_{0}}{4\pi {\left\|r\right\|}^{4}}\left[\frac{Mr^{T}}{\left\|r\right\|}+\frac{rM^{T}}{\left\|r\right\|}-\left(\frac{5{rr}^{T}}{{\left\|r\right\|}^{2}}-I\right)\frac{M^{T}r}{\left\|r\right\|}\right]m$$
where $\mu_{0}=4\pi \times 10^{-7}\;Tm/A$ represents the vacuum permeability, **r** is the vector pointing from the center of the magnet to the controlled device, **I** denotes the unit matrix, and **M** represents the magnetic moment of the dipole source. The force **F** possesses the ability to attract the robot towards the magnetic source of the driver or repel the robot away from the magnetic source of the driver, as illustrated in Figure 3a(ii). This is also the reason for the interaction between the OISR and the magnetic source, which results in the adsorption or detachment of the suction cup from the surface.

The OISR can be attracted under external magnetic gradient forces, causing the internal suction cup to be compressed longitudinally and a portion of the internal air to be expelled, resulting in a pressure difference between the inside and outside. After removing the external gradient magnetic field, the constricted orifice tightens, allowing the OISR to generate stable suction force, as shown in Figure 3b. The theoretical suction force of the suction cup $F_{suc}$ can be expressed as follows [48]:(4)$$F_{suc}=P\left(1-\frac{V_{min}}{V_{0}}\right)A$$
where P represents atmospheric pressure, A denotes the base area of the suction cup, $V_{min}$ represents the volume of air inside the suction cup in the unloaded state, and $V_{0}$ corresponds to the volume of air inside the suction cup before separation.

### 2.4. Driver Design and Magnetic Field Simulation

Compared to electromagnetic systems, permanent magnets can generate strong magnetic fields. The strength and distribution of the magnetic field depend on the size and the shape of the magnet [49]. The dipole model is widely used as it provides a convenient analytical expression for calculations. For a movable permanent-magnet driving device, it can be approximated as a point dipole source located at the center of the magnet’s volume. The magnetic field **B**(**p**) produced by the dipole at position **p** is given by:(5)$$B(p)=\frac{\mu_{0}}{4\pi {\left\|r\right\|}^{3}}\left(\frac{3{rr}^{T}}{{\left\|r\right\|}^{2}}-I\right)M$$

To better control the two motion modes of the OISR, rolling and quick scrolling, we have designed three simple, cost-effective, and easy-to-operate permanent magnet control systems. As shown in Figure 3c, we used a single-axis slide table to move a permanent magnet (N45) with a bottom surface diameter of 50 mm and a height of 90 mm in space to achieve quick scrolling of the OISR. This geometric size was chosen to maintain the magnetic field strength while ensuring a relatively smooth variation of the magnetic field in space, which is conducive to the OISR generating a relatively stable quick scrolling motion. The simulated spatial distribution of the magnetic field is shown in Appendix A. For the rolling motion of the OISR, a stepper motor drives a cubic permanent magnet (N45) with a side length of 25 mm for rotation. We need to provide a sufficient magnetic gradient field to achieve effective adsorption/separation of the biomimetic suction cup. Hence, we used a cylindrical permanent magnet (N45) with a bottom surface diameter of 50 mm and a height of 30 mm for close-range attraction. A single-output flat-panel industrial power supply (MS-500-0-24, MENG WELL, Guangzhou, China) powered the stepper motor driver for motor rotation. As shown in Figure 3d, to visually demonstrate the spatial distribution of the magnetic field for different shapes of permanent magnet drivers, we conducted simulations (COMSOL Multiphysics 6.0, COMSOL, Stockholm, Sweden) of the magnetic fields generated by the permanent-magnet drivers during quick scrolling, rolling, and adsorption/separation. The magnetic field of the permanent-magnet driver exhibits a more drastic spatial variation when the distance is closer. For example, the rolling driver can have a magnetic field strength of about 400 mT at the surface, and when it reaches 10 mm, the magnetic field strength is reduced by 70%.

## 3. Results and Discussion

### 3.1. Assessment of Sucker Suction for OISR

When the suction cup is subjected to tensile forces in the expected direction, the forces cause deformation of the suction cup. The contact area between the suction cup and the surface is pulled inward towards the center of the suction cup mouth. As the tensile force increases, the contact area exhibits a sliding tendency on the base surface. When the suction cup is pulled in the shear direction, the deformation of the suction cup leads to deformation of the contact region, creating vertical pressure on the substrate at the edge of the suction cup, resulting in detachment between the suction cup and the substrate. In the case of a dry surface, the sealing of the contact area is poor, making it prone to air leakage and subsequent detachment of the suction cup [48]. However, if the surface is wet, even when there is sliding or deformation in the contact area, a liquid film in the contact area can prevent air leakage. The presence of GI mucosa can provide better sealing for the suction cup, resulting in stronger adsorption.

By utilizing magnetic gradient force attraction, the suction cup on the OISR evacuates the air from the cup cavity, creating an internal and external pressure difference, thus generating adsorption force. Different magnitudes of magnetic gradient forces result in varying levels of adsorption. The Young’s modulus of gastric wall tissue ranges from 102–103 kPa [50]. We employed a flat surface molded from silicone rubber (Dragon Skin FX-Pro, Smooth-On, Macungie, PA, USA) to simulate gastric wall tissue with a Young’s modulus of 0.25 MPa. The magnet providing the magnetic gradient force was placed at different distances from the suction cup base and then retracted to generate varying suction levels. A force sensor (HP-3, HENDPI, Leqing, China) was attached to measure the peak peeling forces in the normal and shear directions of the OISR, with multiple readings recorded. Under conditions of manual pressing, a permanent magnet at distances of 2 mm, 5 mm, and 10 mm from the suction cup base, and no preload, the peeling resistance capability of the OISR for the normal and shear directions on the simulated gastric tissue surface are shown in Figure 3e. The peeling resistance capability in the normal direction and shear direction achieved by retracting the magnet after adsorption at a distance of 2 mm is approximately 60% and 50%, respectively, of the values achieved by manual pressing. When the suction cup adheres to a surface with a certain curvature, the anti-detachment capability of the OISR is slightly weakened but can still produce adsorption. If a significant opposing repulsion force is applied, the suction cup can directly detach from the simulated gastric wall.

### 3.2. Kinematic Analysis of OISR

Quick scrolling is primarily achieved through the directed attraction provided by the radial magnetic gradient generated by the driver. The magnetic field on the half-axis vertical plane of the cylindrical permanent magnet forms a uniform magnetic field, with the magnetic gradient increasing as it approaches the axis. Therefore, the OISR can achieve quick scrolling at an appropriate position on the axis plane of the cylindrical permanent magnet driver. Suppose there is a slight deviation in position. The reason for the quick scrolling is that it is directly propelled by the magnetic gradient force, making the process highly efficient. In theory, as long as the OISR is within an appropriate controlled space, quick scrolling can be triggered instantly. We measured the drivable range of the quick scrolling motion of the OISR on a plane at different heights relative to the permanent-magnet driver. Specifically, we placed the OISR in a standing position on the plane, either on one side or directly above the permanent magnet driver, and slowly approached the OISR with a single-axis sliding stage until it started to slide. The position at which sliding occurred was recorded as the maximum drivable distance (with the reference plane being the one perpendicular to the center axis of the permanent-magnet driver and the ground). Similarly, we slowly moved the single-axis sliding stage away from the OISR until it slid, recording the position as the minimum drivable distance. As shown in Figure 4a, we can approximate the maximum drivable height of the OISR to be 60 mm, and the smaller the height, the more extensive the drivable range. Additionally, the OISR exhibits different types of rolling depending on the driving speed of the stage. We demonstrated instantaneous scrolling when the permanent-magnet driver propelled the OISR at a speed of 40 mm/s and following scrolling when driven at a speed of 20 mm/s, as shown in Figure 5a and Appendix A. If the OISR is tilted forward or backward, the axial magnetic torque can be balanced with the axial friction to maintain a “standing” state, as shown in the force analysis results in Figure 5b(i).

Rolling motion is primarily generated by rotating the driver’s magnetic moment, and the OISR’s magnetic moment rotates along with the external magnetic field. Due to the strong gradient field of the permanent-magnet driver, the OISR can perform rolling on inclined surfaces and even achieve complete anti-gravity rolling when it is close to the driver. As shown in Figure 5b(ii), we take planar rolling as an example and conduct a force analysis on the OISR. During the driving process, the translational speed of the single-axis slide table and the rotational speed of the permanent magnet should follow a reasonable mathematical relationship. If the translational speed of the single-axis slide table is too high and the rotational speed of the permanent magnet is too low, the OISR may not complete the rolling motion, and its rolling speed cannot keep up with the translational speed of the single-axis slide table. If the translational speed of the single-axis slide table is too low and the rotational speed of the permanent magnet is too high, the OISR’s rolling frequency may be too high, and the strong magnetic gradient may pull the OISR back, causing it to “roll on the spot”. Theoretically, the OISR advances approximately 12.12 mm + 5 mm = 17.12 mm for every 180° of rolling. Therefore, for every revolution of the permanent-magnet driver, the single-axis slide table should move forward by 34.24 mm. This proportional relationship can fluctuate within a certain range without affecting the stability of the motion. Additionally, the rotation direction of the permanent magnet should be such that the rolling direction of the OISR aligns with the direction of motion of the single-axis slide table. Rolling motion has four modes: flat rolling, anti-gravity rolling, tumble climbing, and tumble swimming. We focused on studying the relevant motion parameters of flat rolling and tested the feasibility of anti-gravity rolling, tumble climbing, and tumble swimming. As shown in Figure 4b and Appendix A, by measuring the stable drivable maximum height (the shortest distance between the rotational axis of the permanent magnet driver and the motion plane of the OISR) of flat rolling at different tilt angles under a low rotational speed of 0.625 r/s, it was observed that the drivable height of the OISR gradually decreases with an increasing tilt angle (with a 95% confidence interval for the error bars). The maximum drivable height on a horizontal plane is 95 mm, and on a vertical plane, it is 41.67 mm. As shown in Appendix A, the OISR can still perform stable rolling under anti-gravity conditions, but the drivable distance is limited to only 28 mm. The drivable high degree of the OISR is mainly related to the size of the external magnetic field of the driver. Our cubic permanent magnet (N45) with a side length of 25 mm is designed to respond to the maximum distance that the OISR can respond to under more demanding driving conditions. Combined with the simulation results of the magnetic field, we can effectively infer the external magnetic field required for the robot to move in different environmental conditions. In practical clinical applications, a more costly MRI with a stronger magnetic field can often be used to control the robot in vivo. The farthest drivable distance of the OISR under different heights (the shortest distance between the rotational axis of the permanent-magnet driver and the motion plane of the OISR) and different rotational speeds of the driver (5 r/s, 2.5 r/s, 1.25 r/s, and 0.625 r/s) is shown in Figure 4c. When the distance is very close, the OISR is pulled back due to the excessive gradient force, resulting in a small drivable range. At a distance of approximately 45 mm, where the control effect of the gradient magnetic field weakens, the OISR can move to a location 110 mm away. As the distance increases, the controllable range gradually decreases due to the weakening of the magnetic field. When the distance reaches 80 mm, the gradient field has weakened considerably, and the OISR will oscillate between the controllable edge of the gradient field and a certain position. Higher rotational speeds lead to poorer stability of the motion but faster responses to external conditions, allowing for immediate adjustment of the motion direction and faster movement speed after being disturbed. However, when the rotational speed is too high, the OISR may enter the next motion cycle before completing a whole roll, resulting in the inability to generate effective rolling. As shown in Figure 4d, by observing the motion states of the OISR at different heights and different rotational speeds of the driver, the motion states of the OISR can be artificially divided into four categories: rolling on the spot, rolling back and forth in a certain zone, rolling in a certain zone while occasionally pulling back, and rolling somewhere and stopping.

The two motion modes have different advantages in different application scenarios. Quick scrolling has the characteristics of a fast response speed and a small contact area with the ground, and the drug inside the suction cup does not easily leak. Therefore, by using the quick scrolling method, the OISR can realize the rapid transportation of drugs. However, its controllability is poor, so it needs to be realized under ideal environmental conditions. Rolling is characterized by stable motion and good controllability. Therefore, by rolling, the OISR is able to accurately reach the target position in complex environments (e.g., in the gastrointestinal tract, where there are folds or large amounts of liquid). However, its response time is slow, and it may cause drug leakage.

For the tumble climbing motion, we placed a three-level step on a plane with a height of 40 mm. Due to specific folds on the surface of the gastric tissue, the OISR needs to climb over the steps [51]. Therefore, we set the height of the first step to 10 mm, the height of the second step to 8 mm, and the height of the third step to 6 mm. As shown in Figure 5c, when the rotational speed of the permanent-magnet driver is relatively high (2 r/s) but the sliding stage’s movement speed is low (5 mm/s), the OISR can adaptively perform step climbing actions. Even if there are situations where the OISR fails to grasp the edge of the step and complete the climbing action, it can perform secondary climbing by self-adjusting when the magnetic gradient force in the vertical direction decreases. Ultimately, it can gradually climb to the top plane, as shown in Appendix A.

For tumble swimming, we controlled the motion of the OISR by rotating the permanent magnet at a fixed position. The main purpose was to investigate whether the OISR can overcome gravity and magnetic gradient attraction with the assistance of the liquid reaction force and buoyancy assistance to accomplish a complete upstream and return motion. As shown in Figure 5d and Appendix A, the rolling OISR can effectively resist the influence of gravity and move upstream with the thrust of the water and buoyancy assistance. Moreover, after struggling and rolling several turns, it can also overcome the influence of magnetic gradient attraction and reverse in the direction of the anti-gradient force. However, when the rotational speed of the permanent magnet is too low, the rolling speed of the OISR also slows down synchronously. The slow water flow provides limited reactive power to the OISR, limiting its accumulated energy and its ability to overcome the effects of gravity and magnetic gradient attraction. Therefore, we used a permanent-magnet driver with a rotational speed of 5 r/s to drive the OISR to achieve tumble swimming motion.

### 3.3. In Vivo Application of the OISR

#### 3.3.1. Capsule Guidance In Vivo and OISR Release

To better demonstrate the entire drug delivery process, we encapsulated the OISR in a capsule (Ibuprofen Sustained Release Capsules, GSK, Brentford, UK) with a length of 17 mm and a diameter of 7 mm. This effectively prevents unnecessary leakage of drugs stored inside the suction cup. A 3D-printed curved track with a diameter of 10 mm was created to simulate the tortuous environment inside the intestine. Water pools with depths of 3 mm and 8 mm were placed at the turning points of the track to test the ability of the guiding magnetic field to adjust the capsule’s posture when it is in a levitated floating state and a fully floating state, respectively. As shown in Figure 6a, even in narrow bends, the capsule containing the OISR can be easily dragged by the external guiding magnetic field. However, when the capsule is floating, it requires more time to adjust its posture. The dissolution process of the capsule shell in simulated gastric fluid (pH = 1.5) is depicted in Figure 6b and Appendix A. During the dissolution process, the volume of the capsule expands and the shell becomes softer. At 25 min of dissolution, we flipped the capsule. The capsule stops expanding after 1 h of dissolution. At this point, the capsule can be dragged to the appropriate release location, and the internal OISR can be driven to perform a rolling motion. The faster the rolling speed, the easier it is for the OISR to break free from the completely softened capsule shell. At a rolling speed of 2.5 r/s, the OISR broke free from the constraints of the capsule shell after 1 min. This provides a temporal guideline for controlling the release process of the OISR.

#### 3.3.2. Application of the OISR in a Simulated Stomach

The wide range of drivable distances achievable with the permanent-magnet driver enables the possibility of remote control in more complex environments. As shown in Figure 7a, at different velocities of the permanent-magnet actuator, the OISR can perform both instantaneous scrolling and following scrolling on the wrinkled and uneven surface of the stomach model. However, although this motion pattern exhibits a fast speed response, its controllability is limited, and it can only move near the target location. As depicted in Figure 7b, under the driving of the rotating permanent magnet, the OISR can perform a rolling motion on the surface of the stomach model and precisely reach the target location. Although this motion pattern exhibits a slower speed response, it possesses higher controllability and stability. As shown in Figure 7c, the OISR can stably adhere to the simulated gastric tissue through the attraction of magnetic gradients. When a reverse magnetic gradient field is applied, the OISR will first lift one side of the grappling hooks, and only when the gradient field intensity is further increased will the suction cup completely detach. These processes are visible in Appendix A. As illustrated in Figure 7d and Appendix A, we demonstrate the complete process of continuous gastric drug delivery. By storing blue pigment inside the suction cup to represent the drug, an OISR carrying the drug is fabricated. The specific fabrication method can be found in Appendix A. By observing the entry of the pigment into the drug delivery target site, the feasibility of gastric drug release by the OISR can be determined. The OISR can move to the vicinity of the target location through a rolling motion and further adjust its relative location until the suction cup contacts the drug delivery target site. The preparation method for the drug delivery target site is shown in Appendix A. The repeated attraction by the magnetic gradient field allows the suction cup to adhere to the drug delivery target site stably. After 5 min of sustained drug delivery, applying a reverse gradient field to the OISR detaches it from the drug delivery target. The drug can be partially absorbed by the drug delivery target site, completing the process of drug release. After drug delivery, the OISR can leave the drug delivery target site through a rolling motion. As depicted in Figure 7e and Appendix A, by controlling the OISR to roll towards the swallowed button cell (AG0) in the stomach, its strong magnetic properties can quickly magnetize and stably attract the button cell. The button cell weighs 0.3 g, approximately 1/10th of the weight of the OISR, and the OISR can easily carry it out of the stomach. Even larger and heavier button cells can be successfully removed. This represents a potential application of the OISR.

## 4. Conclusions

We designed and fabricated a magnetically controlled, multimodal, adhesive octopus-inspired soft robot (OISR). The OISR can perform quick scrolling and rolling under the control of an external magnetic field. The biomimetic octopus suction cup within the OISR can achieve adsorption and separation under a gradient magnetic field, generating substantial adsorption force on simulated gastric tissue. We investigated and discussed the motion mechanisms of different motion modes and conducted a detailed study on the relationship between the driver’s rotational speed, maximum driving height, and drivable distance during a flat rolling motion. Furthermore, we explored the implementation conditions for tumble climbing and tumble swimming motions. Additionally, we encapsulated the OISR in a capsule shell, studied the guidance and dissolution processes of the capsule in the body, and demonstrated methods for the OISR to be released from the capsule. Lastly, we investigated the OISR’s motion, adsorption, and separation processes within the stomach and conducted experiments on continuous drug delivery and removing swallowed button cells from the stomach’s interior. The intense magnetism and adhesiveness of the OISR provide a more versatile framework for developing intrabody medical robots. Its multimodal locomotion can further enhance the adaptability of medical robots to complex intrabody environments. This signifies that this research extends the delivery methods and motion patterns of novel intrabody medical robots. In the future, we plan to develop a magnetic localization system for the OISR and further enhance its environmental perception capabilities. By optimizing the design of the robot and centralizing the structure of each part, the stability of the rapid rolling motion and the response speed and movement range of the tumbling motion will be improved. By storing a certain amount of a drug in the suction cup and complementing the interplay of multiple motion modes, we also hope to further realize the task of the OISR reaching different locations in the GI system to complete multiple drug administrations in one go.

## Figures and Tables

**Figure 1 biomimetics-09-00340-f001:**
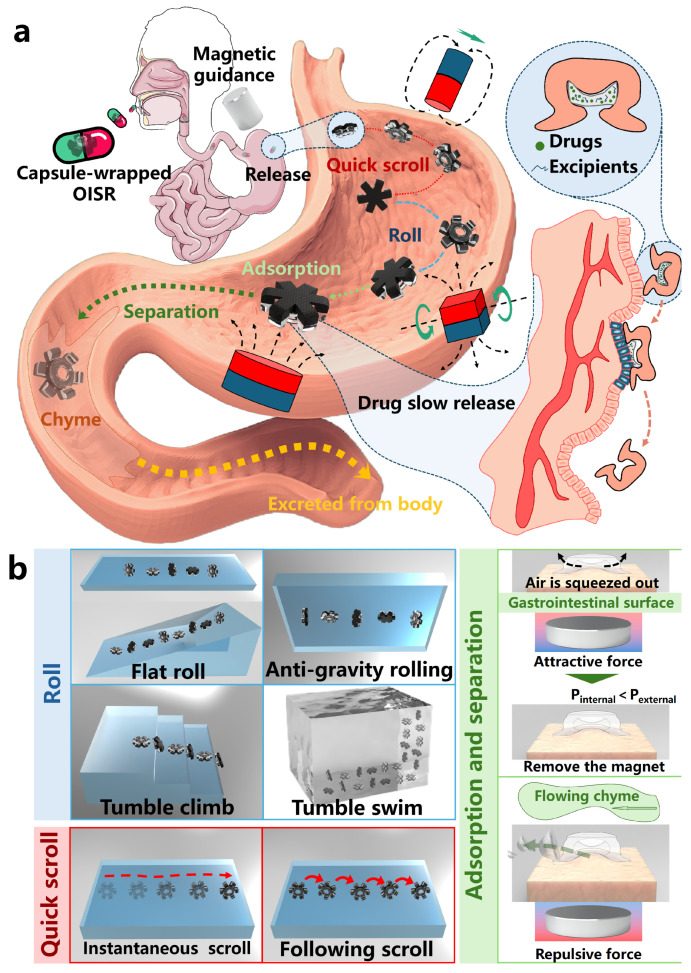
Realization of slow drug release and multimodal motion of OISR. (**a**) The OISR’s suction cup stores the drug internally and moves to the targeted drug delivery site through quick scrolling and rolling motions. The suction cup stably adheres to the drug delivery site through the adhesive force generated by the magnetic gradient, thus achieving slow drug release in the gastric tissue. After drug delivery is completed, the OISR detaches from the gastric tissue and passes into the small intestine along with the process of gastric digestion. (**b**) The OISR exhibits rolling, quick scrolling, adsorption, and separation motions.

**Figure 2 biomimetics-09-00340-f002:**
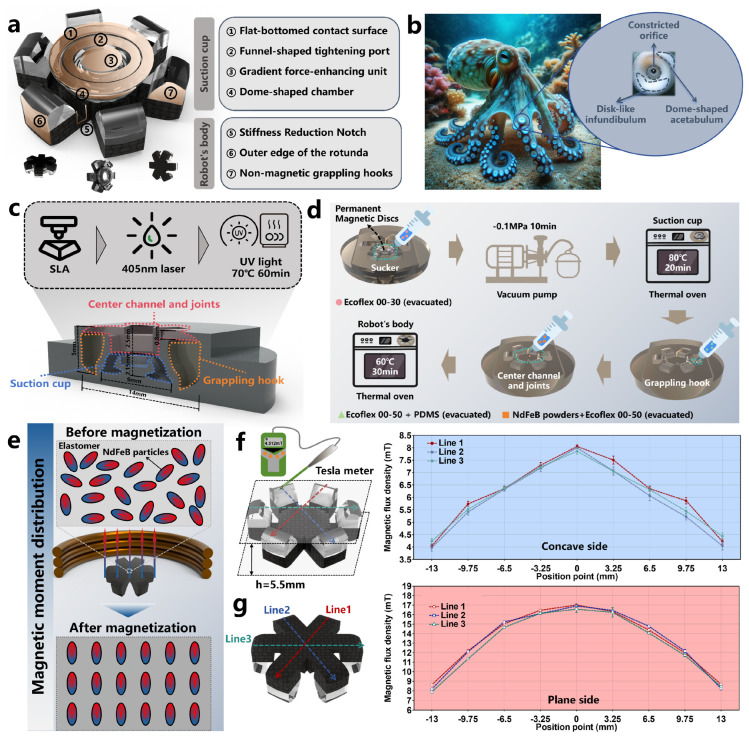
Design, fabrication, and evaluation of OISR. (**a**) Structural composition of OISR. (**b**) Structural characteristics of suckers on octopus tentacles. (**c**) Preparation methods, dimensions, and partitioning of inverse mold. (**d**) Stepwise filling process for robot fabrication. (**e**) Changes in internal magnetic domains of OISR before and after magnetization. (**f**) Distribution of magnetic flux on the concave side. (**g**) Distribution of magnetic flux on the planar side.

**Figure 3 biomimetics-09-00340-f003:**
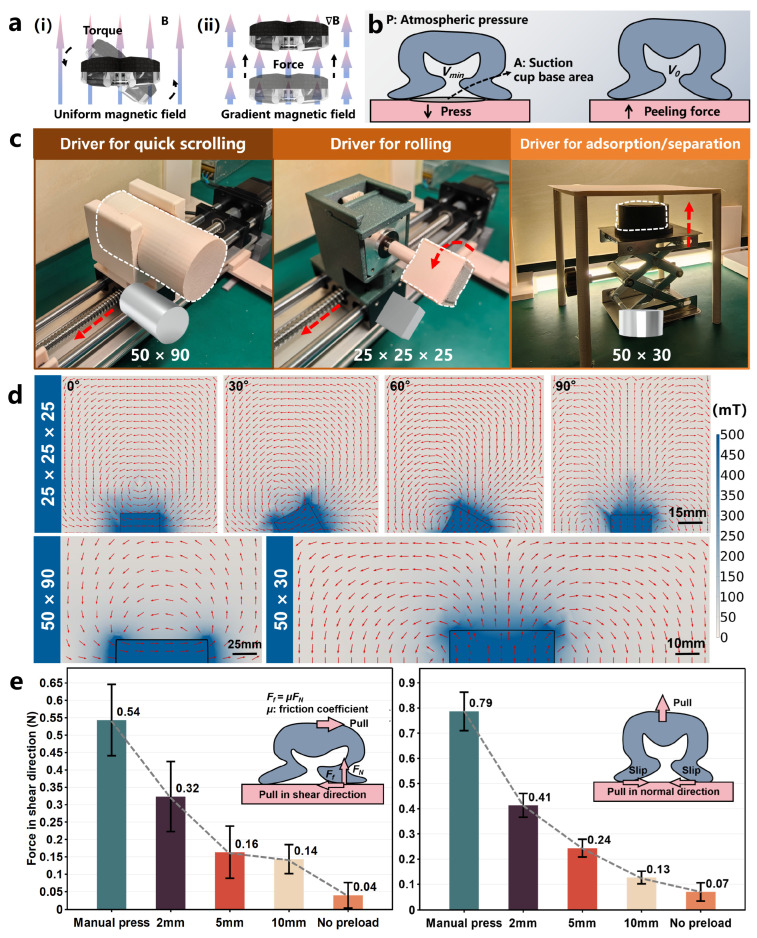
Different driver designs and magnetic field simulations as well as the peel resistance capability of the OISR. (**a**) (**i**): Effects of magnetic torque on the OISR. (**ii**): Effects of magnetic gradient force on the OISR. (**b**) Principle of adsorption force generation by the suction cup. (**c**) The permanent-magnet drivers for rolling, quick scrolling, and adsorption/separation. (**d**) The simulated magnetic field results for the three types of permanent-magnet drivers. (**e**) The peel resistance capability of the OISR in the normal and shear directions, comparing the cases of manual pressing, a permanent magnet at distances of 2 mm, 5 mm, and 10 mm from the suction cup base, and no preload.

**Figure 4 biomimetics-09-00340-f004:**
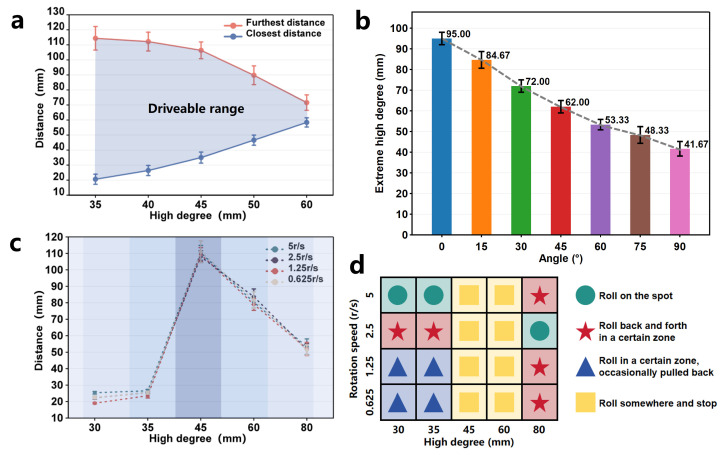
The motion analysis of quick scrolling and rolling of the OISR. (**a**) The drivable distance range for quick scrolling on a plane under different driving heights. (**b**) The maximum drivable height for flat rolling at different tilt angles. (**c**) The farthest drivable distance for flat rolling under different driving heights. (**d**) The four motion states of the OISR: rolling on the spot, rolling back and forth in a certain zone, rolling in a certain zone while occasionally pulling back, and rolling somewhere and stopping.

**Figure 5 biomimetics-09-00340-f005:**
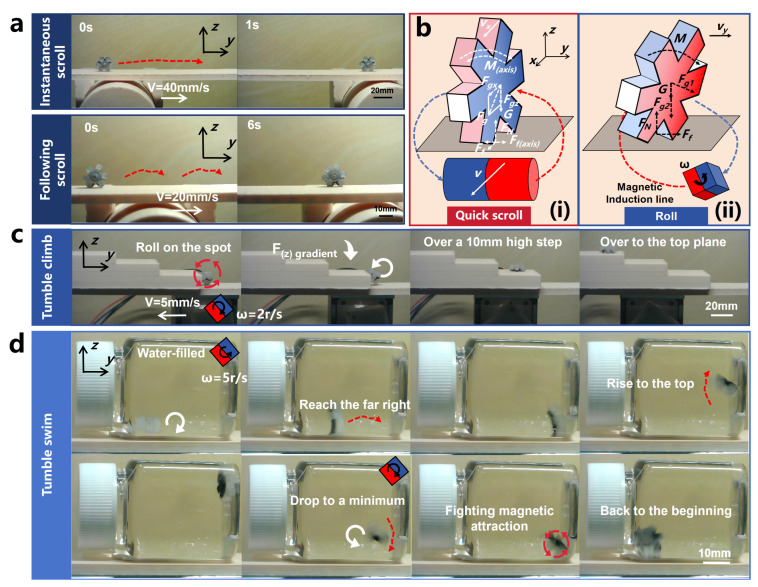
Instantaneous scrolling, following scrolling, tumble climbing, tumble swimming courses of motion, as well as force analyses of quick scrolling and rolling (The direction of the red dashed arrow represents the direction of OISR motion, and the length represents the approximate distance traveled by the OISR for one motion cycle unit. The direction of the red cyclic arrow represents the roll direction of the OISR’s in-place roll. The direction of the white linear arrow represents the direction of motion of the driver. The direction of the white rotary arrow represents the successful completion of a rolling movement of the OISR). (**a**) Two modes of quick scrolling: instantaneous scrolling and following scrolling. (**b**) (**i**): Force analysis during the process of quick scrolling. (**ii**): Force analysis during the process of rolling. (**c**) The process of tumble climbing to the top plane. (**d**) The upstream and return processes of tumble swimming.

**Figure 6 biomimetics-09-00340-f006:**
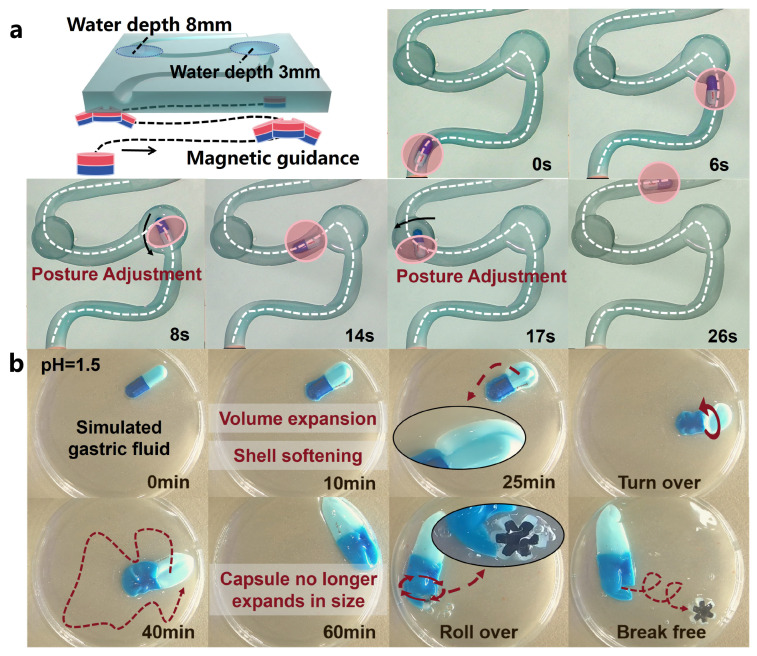
The guidance of the capsule inside the body and the release process of the OISR. (**a**) The capsule is guided through the simulated GI tract under an external magnetic field. (**b**) After the capsule dissolves in the simulated gastric fluid, the OISR is released through rolling.

**Figure 7 biomimetics-09-00340-f007:**
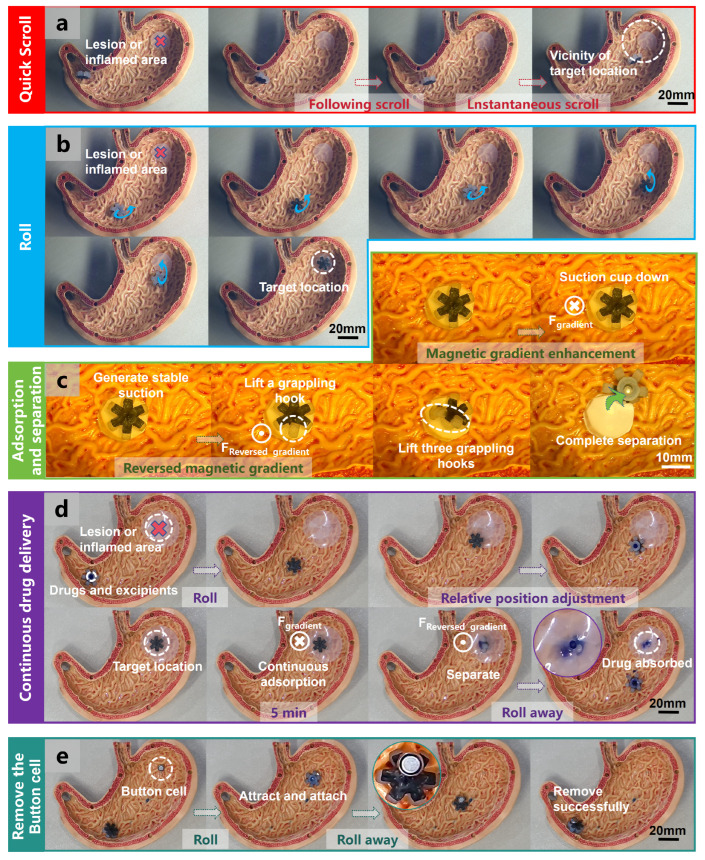
Quick scrolling, rolling, adsorption and separation, continuous drug delivery, and removing the button cell by the OISR in a simulated stomach environment. (**a**) The OISR quick scrolls inside the simulated stomach towards the vicinity of the target location. (**b**) The OISR rolls inside the simulated stomach to reach the target location. (**c**) The adsorption and separation of the OISR on the inner wall tissue of the simulated stomach. (**d**) The process of the OISR reaching the drug delivery target site and achieving continuous drug delivery. (**e**) The process of magnetizing, adhering to, and removing a swallowed button cell inside the stomach by the OISR.

## Data Availability

All data needed to evaluate the conclusions in the paper are present in the paper and/or the Appendix A.

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
