# Peer review of "Octopus-Inspired Soft Robot for Slow Drug Release"

_biomimetics, 2024, doi:10.3390/biomimetics9060340_

Round 1

Reviewer 1 Report

Comments and Suggestions for Authors

The paper includes a well-documented state-of-the-art review section on octopus-inspired soft robots. Also, the research presented in the paper has led to some interesting and useful results (design and fabrication of a magnetically controlled, multimodal, adhesive octopus-inspired soft robot) with good potential for use in on continuous drug delivery. However, I find that the way the figures are grouped into groups of at least 5 graphic elements to a figure (a to e) makes the work hard to read and follow. Also, figures 2 to 5 inclusive appear long before they are mentioned in the text.

Reviewer 2 Report

Comments and Suggestions for Authors

The paper presents design of a robot that resembles an octopus sucker. The robot can potentially deliver drugs to a target area or drag a small foreign body out of gastrointestinal tract. Fabrication of a robot is given in detail. Movable devices are constructed that produce the electromagnetic field needed to control the robot. Experiments are conducted allowing to assess the peel resistance capability of the robot, showing the process of delivering the robot inside a solvable capsule, showing different types of motion of the robot that moves in a stomach model or in test conditions. The paper can be accepted after minor revision.

From the paper it follows that maximum working distance between the robot and the controlling source of electromagnetic field is less than 8cm. It is not clear how is it possible to reach such proximity of the controlling device when using the robot inside a gastrointestinal tract.

As far as the robot’s motion is fully induced from the outside, how is it possible to design the control, in particular, how is it possible to define what type of motion is needed in certain moment of time?

Could you please describe the stomach model (Section 3.3.2) in more details (what materials are used, what are the geometry/adhesion properties, etc.).

Does the proposed robot design have any other potential applications?

It is not clear if the formulas (1)-(5) are somehow used in the research.

The quality of text in figures is poor. It is hardly readable when printed. Zooming in digital form does not help much either. The text should be converted to vector form or dpi should be increased.

Figures 1, 4: “Lnstantaneous“ must be a misspell of “Instantaneous“

Figure 4b: “dgree” – must be a typo.
